# Integrative Care for Challenging Behaviors in People with Intellectual Disabilities to Reduce Challenging Behaviors and Inappropriate Psychotropic Drug Prescribing Compared with Care as Usual: A Cluster-Randomized Trial

**DOI:** 10.3390/ijerph21070950

**Published:** 2024-07-20

**Authors:** Gerda de Kuijper, Josien Jonker, Karlijn Kouwer, Pieter J. Hoekstra, Annelies de Bildt

**Affiliations:** 1Department of Child and Adolescent Psychiatry, University of Groningen, University Medical Center Groningen, Hanzeplein 1, 9713 GZ Groningen, The Netherlands; josien.jonker@ggzdrenthe.nl (J.J.); p.hoekstra@accare.nl (P.J.H.);; 2Mental Healthcare Drenthe, Department Centre for intellectual Disability and Mental Health, Middenweg 19, 9404 LL Assen, The Netherlands; 3Department of Biological and Medical Psychology, University of Bergen, Jonas Lies Vei 91, 5009 Bergen, Norway; 4Accare Child Study Center, Lübeckweg 2, 9723 HE Groningen, The Netherlands

**Keywords:** integrative care, challenging behavior, intellectual disability, psychotropic drugs, off-label prescribing

## Abstract

People with intellectual disabilities (IDs) often present with challenging behaviors (CBs) mostly due to inappropriate environments and mental and physical disorders. Integrative care is recommended to address CBs. However, in clinical practice, psychotropic drugs are often prescribed off-label for CBs, although the effectiveness is unclear, and side effects frequently occur. We conducted a cluster-randomized controlled study to investigate the effect of integrative care provided by a collaboration of an ID specialized mental healthcare team and participants’ own ID service providers’ care team on reducing CBs and inappropriate off-label psychotropic drug prescriptions compared with care as usual. Participants (*N* = 33, aged 19–81 years) had a moderate, severe, or profound intellectual disability and used off-label psychotropic drugs. The primary outcome measures were the Aberrant Behavior Checklist and the total dose of psychotropic drug prescriptions. At the study endpoint of 40 weeks, we found no effect of the intervention on the total ABC score and on the total dose of psychotropic drug prescriptions. In the intervention group, however, the psychotropic drug dose decreased significantly, while CBs did not change. The small sample size and not-completed interventions due to organizational problems may have affected our findings. This study illustrates the difficulties in the implementation of integrative care.

## 1. Introduction

Many individuals with intellectual disabilities show challenging behaviors (also called behaviors of concern), especially in cases of more severe intellectual disability, communication problems and autism [1,2]. The prevalence of challenging behaviors in people with intellectual disabilities varies from 18% in total population studies [1] to 85% in populations of people with severe and multiple disabilities in residential facilities [3]. Challenging behaviors are defined as behaviors of such intensity, frequency, or duration that the physical safety of the person or others is likely to be placed in serious jeopardy or as behaviors that are likely to seriously limit the use of ordinary community facilities or result in the person being denied access to those facilities [4]. These behaviors may include aggression (towards self, other persons, or materials), withdrawn and disruptive behaviors, hyperactivity, and irritability [4].

There are many factors underlying challenging behaviors. Mental [5,6] and physical disorders [7,8] may present with challenging behaviors. Additionally, staff factors like the ability to support clients with challenging behaviors and organizational factors like lack of personnel and insufficient support of staff are associated with the onset and maintenance of challenging behaviors [9,10]. The presence of challenging behavior is negatively associated with the quality of life [11] and functioning [12] of people with intellectual disabilities. Furthermore, families and staff may experience a burden while caring for a person who frequently exhibits challenging behavior [13,14]. These negative consequences highlight a need for effective interventions for challenging behaviors.

Given the multiple factors that may underly challenging behaviors, a multidisciplinary and integrative approach is recommended [15]. Treatment should preferably be non-pharmacological, and long-term psychotropic drug prescription should be avoided [15]. Effective non-pharmacological interventions for reducing challenging behaviors are available. These may also increase the mental well-being of caregivers [16,17]. However, their implementation may be challenging [13,18] in part due to setting-related factors, e.g., lack of policies to stimulate the development of staff skills in the treatment of challenging behaviors [13,18,19].

Despite guideline recommendations, psychotropic drugs, especially antipsychotics, are frequently prescribed outside licensed indications (so-called off-label prescriptions) as a treatment for challenging behaviors, mostly based on their flattening and sedative effects [20,21,22] and often for an extended period. Apart from concerns about its limited effectiveness, the side effects of psychotropics may lead to significant health problems [23], which may even result in challenging behaviors [24]. Furthermore, the use of psychotropic medication can have a negative effect on daily functioning [23] and quality of life [11]. Deprescribing psychotropic medication for challenging behaviors is, therefore, a high priority in clinical practice and policy [15]. Yet, discontinuation of such psychotropic medication may be difficult. Resistance of caregivers and staff who do not know how to deal with challenging behaviors in another way is an important barrier. Insufficient multidisciplinary collaboration and lack of knowledge of/experience with non-pharmacological treatments increase such resistance. Insufficient support of staff to discontinue medication may play a role in this, in part due to system-level barriers with regard to supporting, coaching, and educating staff [19,25,26,27].

To improve a multidisciplinary approach for optimal treatment of challenging behaviors, including the replacement of inappropriate psychotropic drug prescriptions by evidence-based non-pharmacological treatments, e.g., positive behavior support, collaboration is needed between care sectors, as is support from specialized care to community care [25,28]. We developed a program where expertise from specialized mental healthcare and intellectual disability care was combined. We aimed to provide a multidisciplinary and integrative approach and facilitate the reduction of inappropriately prescribed psychotropic medication. The program was piloted in 2018, and evaluations revealed that the professionals involved were enthusiastic about the content of the collaboration but were hindered by organizational and financial barriers. Furthermore, professionals indicated that more involvement from the families and legal representatives was needed.

In the present study, we used this multidisciplinary and integrative care program and tried to overcome these barriers by making agreements to facilitate financing and working processes with the participating service providers. We hypothesized that this approach would lead to a decrease in challenging behaviors and a reduction in inappropriate psychotropic drug use compared with care as usual. Care as usual contained the individualized provision of 24 hour care, daily activities, and primary and secondary healthcare, including the availability of specialized intellectual disability healthcare.

## 2. Materials and Methods

### 2.1. Study Design

We set up a cluster randomized controlled trial (RCT/NL7868/International Clinical Trials Registry Platform/UMCG 201900479) in which we compared integrative care with care as usual.

The clusters were care teams for residents living in units with 24/7 care. These teams consisted of 4–10 intellectual disability support professionals for 8–12 clients who were supported by a designated behavioral scientist. Care teams (i.e., clusters) were randomized per service provider to either integrative care or care as usual by blocks in a 1:1 ratio using a computer-generated randomization list. Teams, clinicians, and researchers were not blinded with regard to allocation. The study took place from January 2020 until January 2023.

#### 2.1.1. Recruitment

First, we approached managers, scientific boards, and medical and behavioral staff of intellectual disability service providers in the catchment area of GGZ Drenthe, a specialized mental healthcare center with a department for individuals with an intellectual disability located in the northeast of the Netherlands. Service providers that expressed interest in the study received information materials, followed by a meeting with the managers, boards, and/or staff. When a service provider agreed to participate, their physicians and behavioral scientists selected care teams with one or more eligible potential participants. These care teams were then informed about the background of the study, the data collection and procedures, and the content of the intervention. Information materials and an informed consent letter were provided to the potential study participants and/or their legal representatives of participating teams. This took place in consultation with the potential participants’ intellectual disability physician, who discussed these materials with the legal representative, and, according to participants’ and professionals’ best interest, with the main caregiver and/or behavioral scientist of the potential participant.

#### 2.1.2. Ethics and Consent Statement 

The authors assert that all procedures contributing to this work comply with the ethical standards of the relevant national and institutional committees on human experimentation, i.e., Dutch legal Acts, Data protection rules and Codes of conduct, and with the Helsinki Declaration of 1975, as revised in 2008. All procedures were approved by the Medical Ethics Review Board of the University Medical Center Groningen (METc2019/429) (6th [and last] amendment 23 March 2023).

We composed a Data Safety Monitoring Board (DSMB) consisting of a pharmacist, psychiatrist, and scientist, all with a doctoral degree, and a board of four experts by experience/parents to guard the safety of the study.

Written informed consent was obtained from all study participants and/or their legal representatives.

### 2.2. Participants

Eligible participants were >12 years, had a moderate, severe, or profound intellectual disability and/or additional multiple disabilities (developmental age < 6 years), and used off-label psychotropic medication for behavioral symptoms for more than one year, i.e., antipsychotics (ATC code N05A), antidepressants (ATC code N06A), hypnotics, or sedatives (ATC code N05C) and/or anxiolytics (ATC code N05B). The exclusion criteria were a diagnosis of dementia, a chronic psychotic disorder, schizoaffective disorder, or bipolar disorder type 1, according to DSM 5-TR criteria.

### 2.3. Integrative Care Intervention

The integrative care intervention was provided in collaboration with the participants’ own care team and an intellectual disability specialized mental healthcare team with the involvement of the participants’ families and/or legal representatives. In all cases, a behavioral scientist from the participant’s own care setting was part of the collaboration with the specialized mental healthcare team, and in half of the cases, also a specialist nurse, general practitioner (GP), and/or intellectual disability physician participated.

The intervention was aimed at assessing and addressing participants’ mental, emotional, and physical health and support needs in order to optimize their functioning. It included evidence-based diagnostics and treatments of mental disorders and challenging behaviors, with an emphasis on behavioral assessments, non-pharmacological treatments, and the deprescribing of off-label used psychotropic drugs. A main part of the intervention was to address causative and maintaining factors of challenging behaviors as the basis for the application of behavioral techniques. Attention was also paid to the emotional development of the person to better understand the nature of the challenging behavior and to tailor the daily guidance to the person’s emotional needs. As Barret et al. (2024) showed, this approach can also be helpful in the reduction of inappropriate psychotropic drug prescriptions [29]. The framework for deprescribing antipsychotics developed by Shankar et al. (2019) served as a guideline [30] to optimize psychotropic drug use. This framework consists of seven domains, i.e., physical and mental health, appropriate use of medication, quality of life/daily functioning and participation, severity of challenging behaviors/risk management plan, functional understanding of the behaviors, and capability of the environment [30]. All domains and their mutual relationships were assessed as potential causative or maintaining factors of participants’ challenging behavior and reasons for the prescription of the off-label psychotropic medication. As recommended in this framework, contingency plans in case of (imminent) behavioral deterioration and regular multidisciplinary evaluations with the involvement of all stakeholders were part of the intervention.

The intervention started with a diagnostic phase of 6–8 weeks. This phase consisted of taking the history from participants’ representatives and care professionals, obtaining information from medical records, i.e., collecting the available diagnostic information and applied treatments, observation and interviewing, and, if necessary, additional genetic, medical, or psychological diagnostics to obtain a clear view of the participant’s personality, health and context, and the nature and severity of the challenging behavior. The next step was to write an intervention/treatment plan after 6 weeks at most. The interventions within that integrative treatment plan included additional diagnostics, individual medical, paramedic, pharmacological (including deprescribing), trauma, psychomotor, and behavioral therapy, and other psychosocial, body-oriented, and systemic/environmental interventions. Treatment was based on the outcomes of the diagnostic phase and was applied to the major problems identified by the specialized mental healthcare team during that phase. The last step was to discuss and reach the consent of the participants’ representatives and their own care teams on the plan up to 2 weeks later.

During the subsequent treatment phase of 30–32 weeks, the interventions as proposed in the treatment plan were provided. Depending on the participants’ condition and the number, nature, and urgency of the interventions, the additional diagnostics and treatments could be applied simultaneously or sequentially.

In order to monitor and evaluate the intervention, meetings of the researchers and the members of the intellectual disability specialized mental healthcare team (i.e., a psychiatrist, intellectual disability physicians, psychologists, and specialist nurses) took place every 2–3 months.

Due to the COVID-19 pandemic and the absence of staff, the intervention had to be provided largely by remote consultations. However, in all cases, visits of the nurses and psychiatrist at participants’ home and daycare locations took place at least once for observations of participants and interviews and consultations with the participants’ care professionals. Furthermore, video materials with observations of participants were available for all members of the specialized mental healthcare team.

### 2.4. Care as Usual

The care as usual that was provided to residents of service providers included the provision of daycare activities and healthcare as provided by paramedics, GPs, and specialist nurses, often in consultation with intellectual disability physicians. The healthcare could also include multidisciplinary care on request and/or on a regular base (once or twice a year when discussing the care plans) by the care teams‘ designated behavioral scientist and intellectual disability physician. Also, a consultant psychiatrist could be available. Care as usual did not include specialist mental healthcare assessments and interventions.

### 2.5. Measures and Materials

Data collection took place at baseline and at time points 8, 16, 24, 32, 40 (study endpoint), and 52 weeks (follow-up). Data on participants were provided by the main caregivers, behavioral scientists, and intellectual disability physicians and/or collected from the files.

Baseline data included demographic characteristics, the etiology of the intellectual disability, the presence of chronic physical and mental health conditions as expressed in International Codes of Primary Care (ICPC), the current use of restrictive measures (including the involuntary administration of medication, the application of mechanical and physical restraints, and restrictions in freedom of movement), symptoms of mental disorders (as assessed by the Psychiatric Assessment Schedule Adults with Developmental Disabilities (PAS-ADD) [31], and life events during the previous year, including their potential negative impact on the person (Checklist Life Events) [CLE] [32].

The primary outcome measure was the total score of the Aberrant Behavior Checklist (ABC), a validated behavioral rating scale that is widely used in intellectual disability research and clinical practice [33,34,35]. The ABC has 58 items that could be rated from 0 (absent) to 3 (severe). The other primary outcome measure was the total dose in defined daily dose (DDD) of psychotropic medication. Dosages of prescribed psychotropic medication were extracted from the files at all time points by the research assistant.

Secondary outcome measures were the five subscales of the ABC (i.e., irritability, lethargy, stereotypic behavior, hyperactivity, and inappropriate speech) and the Behavior Problems Inventory Dutch version for people with profound or multiple disabilities (BPI-PIMD) [36]. This validated instrument consists of four subscales that are rated at a severity and frequency scale: self-injury, stereotypy, withdrawn, and aggressive/destructive behavior. On the frequency scale, each item could be rated from 0 to 4 (no occurrence, monthly-weekly, daily, every hour). The severity scale could be rated from 1 to 3 (minor, moderate, or severe consequences). We used only the BPI frequency scores based on the high correlation between the frequency and severity scales [37,38]. In our study sample, Pearson correlations between the severity and frequency scale at baseline were 0.78 (*p* < 0.001), 0.93 (*p* < 0.001), 0.85 (*p* < 0.001), and 0.9 (*p* < 0.00), respectively, for the subscales self-injury, stereotypy, and withdrawn and aggressive behavior. Both behavioral questionnaires were completed by the participants’ main caregivers.

Data on the quality of the use of psychotropic drugs were assessed with the recently developed Tool Appropriate Psychotropic Drug Prescribing (TAPP) (publication in preparation) on seven domains: indication, dosage, duration, duplication, interactions, evaluation of effect, and evaluation of side effects at the following time points: baseline, 40, and 52 weeks.

The type of diagnostic and treatment interventions applied during the intervention period was provided by the physicians or behavioral scientists of participants and/or collected by the researchers from the participants’ files at the study endpoint (40 weeks). We also collected data on (changes in) participants’ health (ICPC) from the files. Main caregivers were questioned about the occurrence of adverse life events. These were categorized as physical, mental, and contextual adverse events.

The occurrence of medication side effects was measured with the Matson Evaluation Drugs Side Effects Scale (MEDS) and Scales for Outcomes in Parkinson’s Disease–Autonomic Dysfunction (SCOPA-AUT). The MEDS is a validated scale measuring the side effects of psychotropic drugs [39,40] in people with intellectual disabilities, which is currently being validated in the Netherlands (EudraCT 2016-002859-19). In the present study, we used the (translated) subscales Central Nervous System (CNS)-general, CNS-Dystonia, CNS-Parkinsonism/Dyskinesia, and CNS-Behavior/Akathisia. The MEDS consists of a severity scale measuring the severity of symptoms in the two previous weeks (0 = absent; mild = 1; severe = 2) and a duration scale (0 ≤ 1 month; 1 = 1–12 months; 2 ≥ 12 months). Duration is only measured when the severity is >1. We used only the MEDS severity scale because, in this study sample, almost all severity ratings were below cut-off.

The SCOPA-AUT is a validated scale for the measurement of autonomous symptoms in Parkinson’s disease [41] but can also be used to measure the side effects of psychotropics [40], as was done in the current study. Regarding symptoms of the autonomic nervous system (SCOPA-AUT), we calculated scores for those who were continent and those who were incontinent. We excluded the questions on sexual function and, in the latter group, also the questions on bladder function.

Both scales were completed in an interview of the participant and main caregiver by the participant’s nurse or a research assistant.

An overview of the schedule of enrolment, interventions, and assessments is provided in the Appendix A. 

### 2.6. Sample Size

We aimed to include 106 participants spread over 42 clusters. This sample provided the power to detect a medium effect size, i.e., a 0.65 SD decrease in the ABC total score in the intervention group at the study endpoint of 40 weeks. The size was based on our previous findings of a mean total ABC score of 41 and an SD of 25 [42], a prevalence of psychotropic drug prescription of 54% in institutional settings [41], a cluster size of 8–12 residents (with five eligible participants), a participation rate of 50%, a power of 0.8, a *p*-value < 0.05, and drop-out rate of 20%. We assumed that we could recruit eight institutions with six–eight participating clusters (with, on average, five eligible persons) each in our catchment area.

### 2.7. Statistical Analyzes

We used SPSS version 26 for the statistical analyses. To investigate differences between the intervention and control group at baseline, study endpoint, and follow-up, Pearson Chi-square tests were used for categorical variables and independent t-tests for continuous variables. Nonparametric tests were used instead of t-tests in case the data were not normally distributed.

Because the mean number of participants by cluster was small (1.4) and contamination between clusters was common (see Results Section 3), we did not take the cluster structure into account.

We used univariate ANCOVA to analyze the relationship between groups (fixed factor: intervention/control) and the outcome measures, with the baseline values of the outcome measures as covariables.

We used paired samples t-test or Related-Samples Wilcoxon Signed Rank test separately in the intervention and control groups to compare the total ABC scores and total DDD between baseline and the study endpoint of 40 weeks and between baseline and follow-up (52 weeks).

In all analyses, we defined a value of *p* < 0.05 as a significant difference.

## 3. Results

### 3.1. Participants

We recruited six care-providing institutions for participation in the study, who subsequently selected 23 participating clusters. From the clusters, the affiliated clinicians selected, in total, 44 eligible participants. For 38 of those 44 eligible participants, informed consent was provided by their legal representatives. Of the 38 participants included in the study, 18 were randomly allocated to the intervention group spread over 11 clusters and 20 to the control group spread over 12 clusters. However, one care provider (with five participants) withdrew from the study before baseline data were collected because of the COVID-19 pandemic. Ultimately, we could analyze the data of 33 participants (15 intervention and 18 control groups) spread over 23 clusters of living facilities in five care providers. Of those 33, three dropped out of the intervention group (two because of fatal illness and one because of a fatal accident; all events were discussed with the DSMB and not related to the intervention), and two stopped the intervention but remained in the study for data collection. See Figure 1 for the flow diagram of participants.

Inclusion was difficult due to the COVID-19 pandemic and the subsequent high workload of personnel, which hindered the recruitment of care providers, the selection of clusters with eligible participants from participating care providers, and the willingness of participating teams to participate with multiple clients. Although we lengthened the inclusion period by 12 months, it was not possible to achieve the aimed sample size. In discussion with the DSMB, we decided to stop further recruitment after having received the last informed consent on 1 October 2021.

### 3.2. Baseline Characteristics

In Table 1, the participant characteristics are shown.

The age of participants ranged from 19 to 81, with a mean of 49 years. Most participants were male, most had a severe intellectual disability, and the etiology of the disability was most often congenital/genetic. The severity of the behavioral symptoms, as measured with the ABC and BPI, varied largely among the participants.

Regarding the presence of life events, no one reported that these had negatively affected a participant.

The presence of chronic conditions was common (82%), with one ICPC diagnosis in half of the sample and two or more in a third.

The majority of psychotropic drug prescriptions were antipsychotics. Known reasons for psychotropic drug prescriptions were often challenging behavior, but most often, reasons were not noted in the record and were unknown to the participants’ caregivers and clinicians. The mean number of prescribed psychotropics (including anti-epileptics) was 2.6 (SD 1.9), with a range of 1–7. The mean number of antipsychotic drug prescriptions by antipsychotic drug users was one, and this was also the case for antidepressants and hypnotics. Regarding anti-epileptics, the mean number was 2.5, and for anxiolytics, this was 1.2. The mean dose of antipsychotic drugs per user was 0.54 DDD, that of antidepressants was 1.1, that of anti-epileptics was 1.6, that of hypnotics was 1.5, and that of anxiolytics was 0.94 DDD.

Most participants experienced one or more neurological side effects, especially akathisia and parkinsonism/dyskinesia.

At baseline, there were no differences between the intervention and control group regarding the behavior as measured with the ABC and BPI, age, sex, level of intellectual disability, etiology, the presence of genetic syndromes, symptoms of mental disorders above the cut-off (as measured with the PAS-ADD Checklist), the number of life events during the foregoing 12 months (as measured with the CLE), the deployment of restrictive measures, and the presence of neurological side effects. The baseline total dose and number of psychotropics were significantly higher in the intervention group compared with the control group.

### 3.3. Interventions

Data on diagnostic and treatment interventions in the control group were not provided by the responsible professionals and were also not noted in the records. Table 2 shows the content of the treatment plans in the intervention group.

Information from the records of the participants in the intervention group indicated that it was not always possible to apply the intervention (i.e., providing integrative care while reducing inappropriate psychotropic drug prescription) as planned regarding the timetable for the different phases, as well as regarding the performing of the proposed interventions. Of all fifteen participants in this group, intervention plans were established but often not within the planned 8 weeks. The mean length of time for diagnostics and the completion of the intervention plans could be calculated for 13 of the 15 participants and ranged from 2 weeks to 19 weeks (mean 8.2, median 7 weeks). During the intervention period, three of the 15 participants died, and two stopped the intervention prematurely. None of the intervention/treatment plans of the remaining 10 participants, notably the non-pharmacological interventions, were finished at the study endpoint or not even at follow-up (only one participant had finished the treatment by then). Moreover, in some cases, interventions were not even started at all, especially behavioral interventions (n = 9). Information from the records showed that appointments for multidisciplinary evaluations were frequently canceled by the participants’ care teams because of day-to-day issues. Also, the fine-tuning of working processes between the care sectors was hindered by the procedural requirements of health insurance companies and by differences in organizational structures.

In our study, we aimed to use the framework for the reduction of inappropriately prescribed psychotropics by Shankar et al. (2019) as a guideline in the development of intervention plans [30]. Diagnostics included, in all cases, an evaluation or update of challenging behavior risk management and/or contingency plans. However, it was often not possible to immediately address all the questions and issues on participants’ needs arising from the six remaining domains (physical and mental health, medication use/appropriate prescribing, daily functioning, functional understanding of the behaviors, and capability of the environment), either diagnostically or as a target for treatment. The treatment plans were especially difficult to realize. Information from the records of participants on reports of multidisciplinary evaluations showed that information and knowledge on appropriate psychotropic drug use, how to address potential behavioral responses to a medication change, and the functional understanding of clients’ challenging behaviors were provided by the mental healthcare team. However, during the three-monthly meetings of the researchers and the specialized mental healthcare team, it became clear that transfer of this knowledge within teams and/or between the living situation setting and the daycare center setting was often lacking, and/or this knowledge was not applied in daily care. Therefore, although relatives and/or other representatives were closely involved in all cases and in treatment evaluations emphasized the application of the proposed contextual interventions, teams were often not able to adhere to the plans.

### 3.4. Effects of the Intervention on Challenging Behaviors and on Psychotropic Drug Dose

The results of the univariate ANCOVA showed that there was no relationship between the allocated treatment group and challenging behaviors as measured at the study endpoint (40 weeks), including ABC total scores, ABC subscale scores, and BPI subscales. Also, there was no relationship between the allocated treatment and the total dose of prescribed psychotropic drugs (in DDD), nor with the total dose of the various groups of psychotropic drugs measured at 40 weeks.

### 3.5. Course of Severity of Behavioral Symptoms and Psychotropic Drug Use in the Intervention and Control Group

Figure 2 shows the course of the means of the ABC total score for the two groups of participants at baseline, during the intervention period, at the study endpoint (40 weeks), and at follow-up (52 weeks). Paired sample t-tests within the two treatment groups separately showed that the within-group differences between baseline and 40 weeks and between baseline and follow-up were not significant in either group.

Figure 3 shows the course of the ABC subscales for the intervention and control group, and Figure 4 shows the course of the BPI/frequency subscales at baseline and at the time points 8, 24, 40, and 52 weeks.

Table 3 shows the changes in psychotropic drug use in the intervention and control group between baseline and study endpoint (40 weeks) and follow-up (52 weeks). Related-Samples Wilcoxon Signed Rank test revealed a significant reduction in total dose of psychotropics between baseline and 40 weeks in the intervention group (standardized test statistic = −2.43, *p* = 0.015). Of note, at baseline, there was a significant difference between the intervention and control group regarding the total number and total dose of psychotropic drugs (standardized test statistic = −2.29, *p* = 0.03; and standardized test statistic = −2.01, *p* = 0.44, respectively).

Unfortunately, data on the quality of psychotropic drug prescriptions as measured with the TAPP were missing. We were not able to fill in the TAPP since data on the seven domains were not systematically documented in the participants’ records.

### 3.6. Comparison between Groups Regarding the Occurrence of Side Effects and Adverse Events

There were no differences between both groups in the presence and severity of side effects as measured with the MEDS and SCOPA-AUT (Mann Whitney-U test) at the study endpoint (40 weeks), the number of physical, mental and/or contextual adverse events (only measured at 40 weeks), and the deployment of the various restrictive measures (only measured at 52 weeks) (both Pearson Chi-square test) between the groups. In both groups, there were also no changes in the presence of comorbid disorders as categorized with ICPC and the occurrence of life events as measured with the CLE between baseline and 52 weeks.

## 4. Discussion

In this randomized controlled study, we compared integrative care for decreasing challenging behaviors and reducing inappropriately prescribed psychotropic drugs of people with moderate, severe, or profound intellectual disabilities to care as usual. We found no effect of integrative care on the severity of challenging behaviors nor on the total dose of prescribed psychotropic drugs. Although the total psychotropic drug dose decreased significantly in the intervention group, based on our study, we cannot conclude whether an integrative approach as undertaken in this study is or is not effective for decreasing challenging behaviors or psychotropic drug dose. The sample size was much smaller than anticipated. We recruited 38 participants and were able to analyze the data of 33 participants, where we expected to include three times as many based on the catchment area (the expected n of eligible clients was 240–320, with a 50% attrition rate). We expected the calculated sample size to be reachable by including eight institutions, with an average of six to eight clusters and about four to six eligible participants per cluster. However, recruitment was heavily hindered by organizational and personnel problems of the service providers and organizational problems resulting from COVID restrictions.

Our study sample consisted of individuals with many comorbidities. Almost half of the study participants had one or two, and one-third had three or more comorbid mental and/or physical disorders; approximately one-third had a genetic syndrome, and over 40% had autism. High mental, neurological, and somatic comorbidities were also found in other studies among people with intellectual disabilities [42,43,44], which may underline the need for multidisciplinary, integrative care, especially in the case of challenging behaviors [15].

It is important to note that in the integrative care group, all participants received at least one additional behavioral, psychiatric, and/or somatic diagnostic assessment, indicating that needs existed that were not met previously and that further assessment was seen as an important step to understand the underlying factors of the challenging behaviors. Unfortunately, we could not compare this to diagnostic and treatment interventions in the control group, as no data were available. Our study seems to corroborate the findings of Rose [9] that people with intellectual disabilities and challenging behaviors form a complex, heterogeneous group with a large variety of health needs that are not always met when providing usual intellectual disability care. However, although the provision of multidisciplinary, integrative care to optimize the assessment and treatment of challenging behaviors is recommended [15], the implementation of this kind of care in clinical and research practice is not obvious [45], as was also seen in our study.

Despite the limitations of the small sample size, insufficient implementation of the intervention, and lack of significant findings, valuable lessons can be learned from the experiences in conducting this study. Providing integrative care to people with challenging behaviors may be considered a complex intervention manifesting at different levels of care provision [19]. During our study, this complexity became very clear.

First, at the team level, it was not always possible to apply the intervention (i.e., providing integrative care while reducing inappropriate psychotropic drug prescription) as planned. Insufficient communication within teams and between the daycare setting and living setting about tasks related to the interventions caused delays in diagnostics and played a role in disagreements within teams regarding the implementation of the treatment plans.

Second, at the organizational level, the differences in working culture and working processes between specialized mental healthcare teams and intellectual disability care also caused delays in the progression of the intervention or even cancellation of interventions. Here, a lack of coordination and tuning in working processes between service providers seemed to be an important barrier. Also, on the organizational level, the capabilities of teams to conduct the proposed interventions were limited, and communication and collaboration among all stakeholders from the different care sectors, including client’s representatives, could not fully be supported and facilitated, leading to problems in implementing the proposed intervention plans.

Third, at the system level, the collaborating care sectors had different financing structures and different perspectives on care provision, which made it difficult to find a way to carry out integrative care, respecting views from both care sectors and without extra costs. Integration of care systems could be stimulated by financing and policy alignment, as well as interagency cooperation and cross-training [46,47]; however, the (cost)effectiveness of this kind of care is yet unclear [27,48].

Strategies to overcome the barriers in the provision of integrative care to people with intellectual disability and challenging behaviors should at least address the three above-mentioned levels. The starting point should be at a conditional level regarding the organization and content of care. To improve these conditions, intersectoral cooperation is needed. For such cooperation, governments and health insurance should facilitate the financing of combining care activities from different care sectors. Additionally, managers and professionals of the different care sectors will have to share the problem ownership and jointly set goals to provide appropriate care for people with complex health needs. Only with cooperation at both levels will it be possible to invest in the training and education of care professionals at work, which is needed to boost their knowledge, capabilities, and work pleasure in providing care to their clients with challenging behaviors. In this way, the quality of cross-cutting healthcare for people with complex care needs will likely improve.

Furthermore, based on these lessons and given the shared need of care providers to provide integrative care to people with intellectual disabilities and challenging behaviors, well-designed studies into the development and effectiveness of such integrative care are needed. For that aim, study designs are needed that are feasible to conduct in the intellectual disability mental healthcare field and that take into account the heterogeneity, multiple problems, and difficulties in recruitment [49], as well as difficulties caused by blinding [50]. Researchers may question whether the use of a randomized control trial design is suitable for investigating complex interventions in the intellectual disability mental healthcare research field. They should seek robust study designs that are acceptable for care professionals and take into account conditions in service providers that make data collection and complex interventions feasible.

### 4.1. Limitations

The main limitation of this randomized controlled study was the small sample size and subsequent lack of power to test our study hypothesis of decreasing challenging behaviors and reducing psychotropic drug use. This may have affected the findings of no change in challenging behaviors as measured with the ABC and in psychotropic drug dose during and after the intervention. In particular, possible effect of the intervention on the reduction of the psychotropic drug dose may have remained undetected due to a lack of power caused by the small number of participants and the incomplete course of many interventions. Another important limitation is the incompleteness of the intervention. An even more thorough preparation with timely and active participation of all stakeholders to conduct a complex intervention trial such as this drug reduction and providing integrative care is necessary [51].

Yet another limitation was the chance of contamination between clusters within a care provider because participants in control and intervention groups sometimes shared GPs, intellectual disability physicians, specialist nurses, and behavioral scientists. Furthermore, data collection was not blinded due to personnel problems. Last, the teams and participants were selected by clinicians of participating organizations, so the results of this study may not be generalizable to other settings, even within the same organization.

### 4.2. Conclusions and Recommendations

From this study, we cannot draw conclusions regarding the effect of providing integrative care on the severity of challenging behaviors of people with intellectual disabilities and on off-label psychotropic drug use.

To improve the knowledge and research in this subject, it is suggested to take into account conditions within the healthcare system and at an organizational and team level that should be met in conducting integrative care in this population with complex health needs. It is imperative that researchers, directors, policymakers, and care professionals collaborate in setting up studies to investigate which kind of care and care systems are needed to address the health and support needs of people with intellectual disabilities and challenging behaviors. When doing so, suitable study designs for complex interventions in the research field should be sought.

## Figures and Tables

**Figure 1 ijerph-21-00950-f001:**
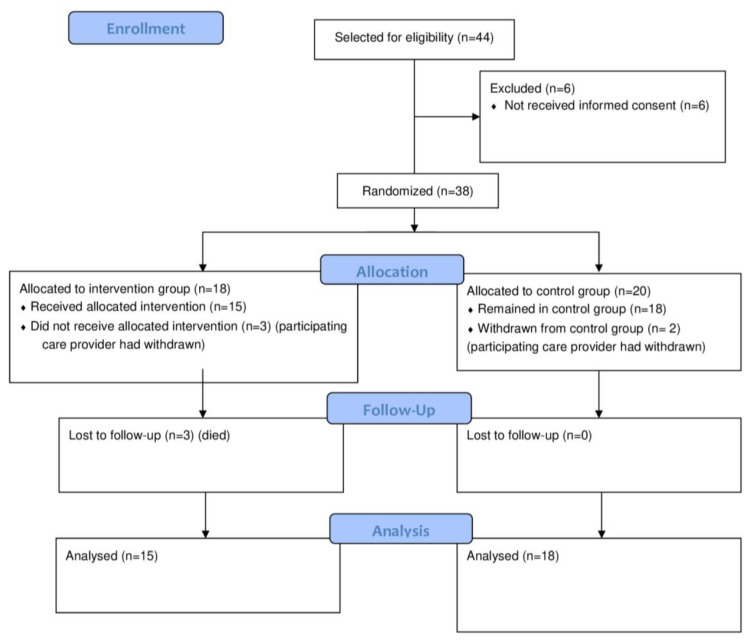
Flow diagram of participants.

**Figure 2 ijerph-21-00950-f002:**
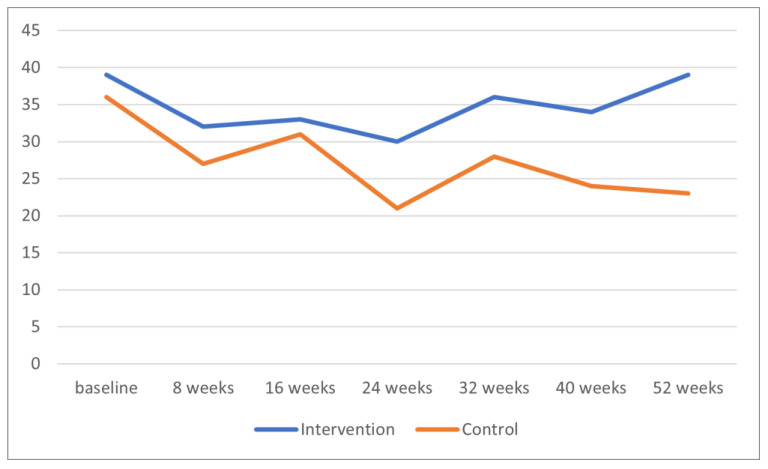
Course of ABC total scores of participants in a randomized controlled trial of integrated care (intervention) versus care as usual (control). ABC = Aberrant Behavior Checklist.

**Figure 3 ijerph-21-00950-f003:**
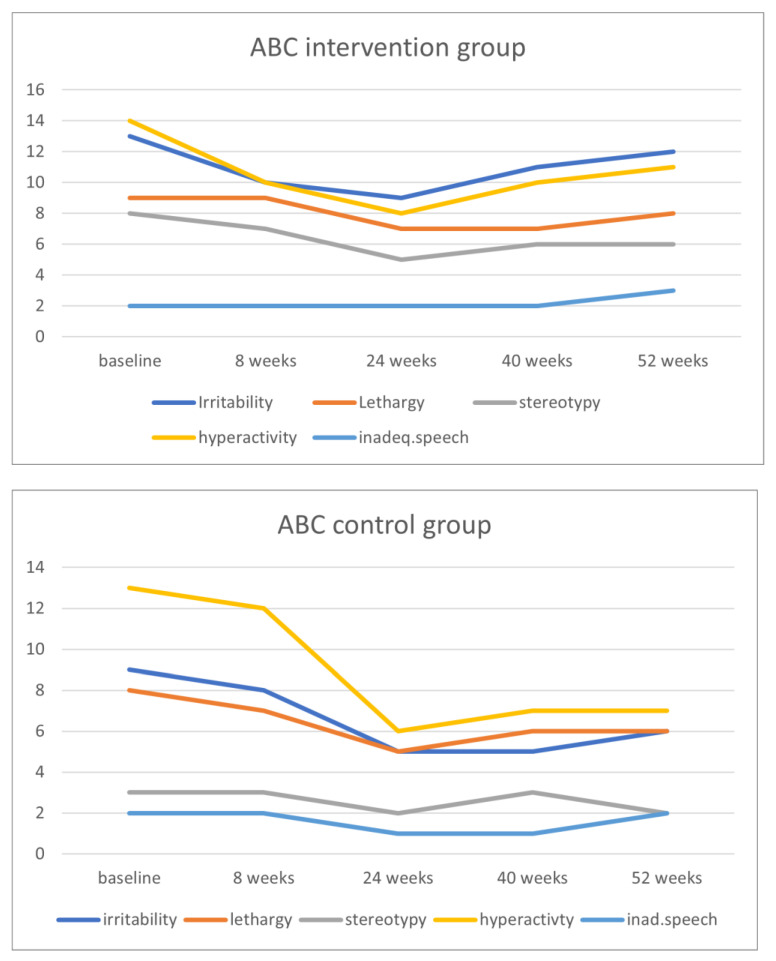
Course of the Aberrant Behavior Checklist (ABC) subscales scores in the intervention and control groups.

**Figure 4 ijerph-21-00950-f004:**
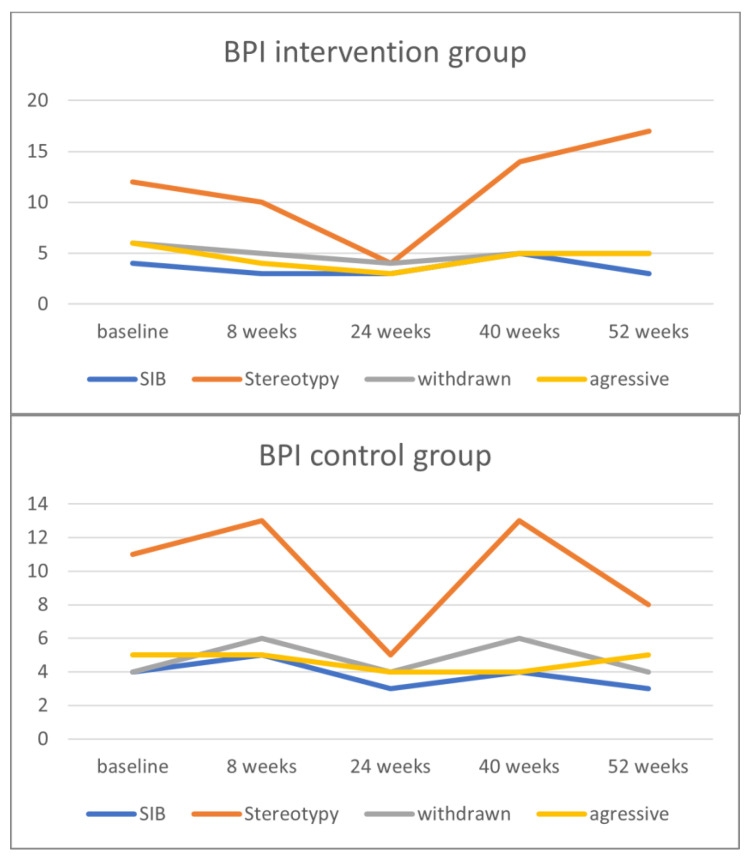
Course of the Behavior Problems Inventory-Profound and Multiple Disabilities (BPI-PIMD) subscale scores in the intervention and control groups. SIB = self-injurious behavior.

**Table 1 ijerph-21-00950-t001:** Baseline participants characteristics (whole sample N = 33).

Participant Characteristics		Mean (SD)/% Total Group (N)	Mean (SD)/% Intervention Group (n)	Mean (SD)/% Control Group (n)
Age in years ^a^ (N = 33)		49 (16.4) (N = 33)	50.6 (15.4) (n = 15)	47.7 (17.6) (n = 18)
Gender (N = 33)	Male	79% (N = 26)	73% (n = 11)	83% (n = 15)
	Female	21% (N = 7)	26% (n = 4)	17% (n = 3)
Level of intellectual disability (N = 33)	Mild	3% (N = 1)	6% (n = 1)	0% (n = 0)
	Moderate	18% (N = 6)	13% (n = 2)	22% (n = 4)
	Severe	61% (N = 20)	60% (n = 9)	61% (n = 11)
	Profound	18% (N = 6)	20% (n = 3)	17% (n = 3)
Etiology (N = 33)	Congenital/genetic	64% (N = 21)	73% (n = 11)	55% (n = 10)
	Perinatal	6% (N = 2)	6% (n = 1)	5% (n = 1)
	Acquired	15% (N = 5)	13% (n = 2)	17% (n = 3)
	Congenital and acquired	15% (N = 5)	6% (n = 1)	22% (n = 4)
Severity of behavioral symptoms ^b^ (N = 33)		(N = 33)	(n = 15)	(n = 18)
	ABC ^b^ total	38.8 (30.3)	42.4 (32.6)	35.8 (29.0)
	Irritability	10.9 (10.5)	12.7 (11.2)	9.4 (10.0)
	Lethargy	8.7 (7.5)	9.2 (9.0)	8.3 (6.1)
	Stereotypy	3.8 (4.9)	4.2 (5.7)	3.4 (4.1)
	Hyperactivity	13.2 (11.2)	14.0 (10.3)	12.6 (12.1)
	Inadequate speech	2.2 (2.8)	2.3 (3.1)	2.1 (2.7)
	BPI ^b,c^			
	Self-injurious behavior	3.6 (3.4)	3.5 (3.5)	3.6 (3.4)
	Stereotypic behavior	11.2 (9.7)	12.3 (11.0)	10.2 (8.8)
	Withdrawn behavior	5.0 (4.3)	5.7 (4.9)	4.4 (3.9)
	Aggressive/destructive behavior	4.9 (5.1)	5.9 (5.9)	4.1 (4.3)
Symptoms of mental disorders ^d^ (N = 33)	Scale organic condition	6% (N = 2)	6% (n = 1)	6% (n = 1)
	Scale affective disorder	6% (N = 2)	0% (n = 0)	6% (n = 2)
	Scale psychotic disorder	18% (N = 6)	29% (n = 3)	17% (n = 3)
Number of life events ^e^ (N = 33)		4.4 (1.8) (N = 33)	3.7 (1.3) (n = 15)	4.9 (2.0) (n = 18)
Presence of physical and mental conditions ^f^ (N = 33)	Gastro-intestinal	15% (N = 5)	13% (n = 2)	17% (n = 3)
	Eye/vision	27% (N = 9)	20% (n = 3)	33% (n = 6)
	Hearing	18% (N = 6)	13% (n = 2)	22% (n = 4)
	Cardiovascular	15% (N = 5)	0% (n = 0)	27% (n = 5)
	Musculoskeletal	15% (N = 5)	13% (n = 2)	17% (n = 3)
	Neurological (including epilepsy)	36% (N = 12)	40% (n = 6)	33% (n = 6)
	Mental (excluding autism)	33% (N = 11)	33% (n = 5)	33% (n = 6)
	Other (each heading code < 15%)	36% (N = 13)	60% (n = 9)	22% (n = 4)
	>2 codes	33% (N = 11)	27% (n = 4)	39% (n = 7)
	1 or 2 codes	49% (N = 16)	60% (n = 9	39% (n = 7)
Other conditions (N = 33)	Autism	42% (N = 14)	47% (n = 7)	39% (n = 7)
	Genetic syndroms	33% (N = 11)	40% (n = 6)	28% (n = 5)
Deployment of restrictive measures (N = 32)	Chemical (PD ^g^ for challenging behavior)	59% (N = 19)	67% (n = 10)	53% (n = 9)
	Physical restriction	22% (N = 7)	27% (n = 4)	18% (n = 3)
	Lock in	6% (N = 2)	6% (n = 1)	6% (n = 1)
Psychotropic drug use (N = 33)	Antipsychotics	88% (N = 29)	93% (n = 14)	83% (n = 15)
	Reason for prescription:			
	Behavior	38% (N = 11)	50% (n = 7)	27% (n = 4)
	Psychosis	3% (N = 1)	7% (n = 1)	0% (n = 0)
	Unknown	59% (N = 17)	43% (n = 6)	73% (n = 11)
	Antidepressants	33% (N = 11)	47% (n = 7)	22% (n = 4)
	Reason for prescription:			
	Behavior	9%/0%/25%	0% (n = 0)	25% (n = 1)
	Depression/anxiety	18% (N = 2)	14% (n = 1)	25% (n = 1)
	Unknown	73%/86%/50%	86% (n = 6)	50% (n = 2)
	Anti-epileptics/mood stabilizers	39% (N = 13)	60% (n = 9)	22% (n = 4)
	Reason for prescription:			
	Epilepsy	46% (N = 6)	44% (n = 4)	50% (n = 2)
	Mood disorder	15% (N = 2)	22% (n = 2)	0% (n = 0)
	Unknown	39% (N = 5)	34% (n = 3)	50% (n = 2)
	Hypnotics/sedatives	15% (N = 5)	20% (n = 3)	11% (n = 2)
	Reason for prescription:			
	Sleep disorder	20% (N = 1)	33% (n = 1)	0% (n = 0)
	Unknown	80% (N = 4)	67% (n = 2)	100% (n = 2)
	Anxiolytics	15% (N = 5)	13% (n = 2)	17% (n = 3)
	Reason for prescription:			
	Behavior	20% (N = 1)	50% (n = 1)	0% (n = 0)
	Unknown	80% (N = 4)	50% (n = 1)	100% (n = 3)
Presence of neurological side effects ^h^	CNS ^i^ (range 0–28)	3.9 (2.4) (N = 26)	4.2 (2.3) (n = 13)	3.6 (2.5) (n = 13)
	Dystonia (range 0–10)	1.45 (0.7) (N = 11)	1.4 (0.5) (n = 5)	1.5 (0.8) (n = 6)
	Parkinsonism/dyskinesia (range 0–28)	3.1 (2.6) (N = 28)	2.9 (2.8) (n = 13)	3.3 (2.2) (n = 15)
	Behavior/akathisia (range 0–16)	4.7 (3.0) (N = 30)	5.3 (2.8) (n = 14)	4.0 (3.2) (n = 16
Presence of autonomic side effects ^j^ (n = 19)	For those continent (range 0–57)	6.0(3.2) (N = 19)	17.9 (6.7) (n = 10)	7.2 (7.9) (n = 9)
(n = 13)	For those incontinent ^k^ (range 0–45)	9.0 (6.8) (N = 13)	5.0 (5.0) (n = 5)	5.9 (2.5) (n = 8)

^a^ Range was 19–81 years of age. ^b^ As measured by the Aberrant Behavior Checklist (ABC) and the Behavior Problems Inventory (BPI). ^c^ Because of the strong correlation between the subscale severity and frequency, only the frequency scale is presented. ^d^ Symptoms of mental disorders were counted if the score was above the cut-off score of the subscales of the Psychiatric Assessment Schedule for Adults with Developmental Disabilities (PAS-ADD). ^e^ Number of life events in the foregoing 12 months as measured by the Checklist Life Events. ^f^ As scored by the heading codes of the International Classification of Primary Care (ICPC). ^g^ Psychotropic drugs (PD). ^h^ As measured by the Matson Evaluation of Drug Side Effects Scale and shows the scores for those participants with scores > 0. ^i^ Central Nervous System (CNS). ^j^ As measured by the Scale Outcomes Parkinson’s Disease—Autonomic Dysfunction (SCOPA-AUT). ^k^ Questions on bladder function excluded.

**Table 2 ijerph-21-00950-t002:** Content of treatment plans of participants (n = 15) in the intervention group of a randomized controlled trial on integrative care versus care as usual for individuals with moderate or severe intellectual disability and challenging behavior.

Additional Diagnostics	Number of Cases	Treatments	Number of Cases
Developmental	4	Psychological/behavioral ^1^	0
Psychiatric	7	Physical/lifestyle	3
Behavioral analyses	9	(De)prescribing psychotropics/ anti-epileptics	14 3
Medical		Team coaching and education, team self-reflection and system therapy	4
-etiological	4		
-epilepsy	3		
-visual function	3		
-other	5		
Sensorimotor processing	3	Adjustment day program and guiding	3
Communication	5	Adjustment communication	4
Contextual			
-team interactions	6		
-daycare/day program	5		

^1^ Treatment proposals in case of completed behavioral analysis; however, participants’ care teams were not able to or did not agree to cooperate in the treatment and/or representatives did not provide consent.

**Table 3 ijerph-21-00950-t003:** Changes in defined daily dose (DDD) (median, interquartile range) and number (median, interquartile range) of psychotropic drugs between baseline/endpoint (40 weeks) and baseline/follow-up (52 weeks) in the intervention and control group of a randomized controlled trial (n = 33) on integrative treatment of challenging behavior.

	Intervention Group (n = 15)	Control Group (n = 18)	Differences between Groups
DDD psychotropics			
Baseline			*p* = 0.04 ^2^
N	15 ^1^	18	
Median (IQR)	2.7 (0.5–3.8)	0.9 (0.2–2.3)	
40 weeks			
N	12 ^1^	16	*p* = 0.35
Median (IQR)	2.0 (0.3–2.8)	1.0 (0.2–1.5)	
52 weeks			*p* = 0.42
N	10	16	
Median (IQR)	1.9 (0.2–2.9)	1.0 (0.2–1.0)	
Total number of psychotropics			
Baseline			*p* = 0.03 ^2^
N	15	18	
Median (IQR)	3.0 (1.0–5.0)	1.0 (1.0–2.3)	
40 weeks			*p* = 0.37
N	12	16	
Median (IQR)	2.5 (1.0–3.8)	1.0 (1.0–2.0)	
52 weeks			*p* = 0.31
N	10	16	
Median (IQR)	2.5 (1.0–3.0)	1.0 (1.0–2.8)	
Changes total dose psychotropics ^3^			
between baseline/40 week:			*p* = 0.3 ^3^
No change	25%	44%	
Decrease ≤ 50%	41%	19%	
Decrease > 50%	17%	12%	
Increase ≤ 50%	17%	6%	
Increase > 50%	0%	19%	
Changes total dose psychotropics ^3^			
between baseline/52 weeks:			*p* = 0.9 ^3^
No change	20%	31%	
Decrease ≤ 50%	20%	6%	
Decrease > 50%	10%	13%	
Increase ≤ 50%	30%	31%	
Increase > 50%	20%	19%	

^1^ Significant difference in the intervention group between baseline and 40 weeks (Related-Samples Wilcoxon Signed Rank test; standardized test statistic = −2.43, *p* = 0.015). ^2^ Significant differences between the intervention and control group (Independent-Samples Whitney U tests (standardized test statistic = −2.01, *p* = 0.04 and standardized test statistic = −2.28, *p* = 0.03, respectively).). ^3^ No significant differences between the groups (Pearson Chi-square tests). IQR = interquartile range

## Data Availability

Data gathered in this study are available on request at Mental Healthcare Drenthe: research.cvbp@ggzdrenthe.

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
