# Peer review of "Integrative Care for Challenging Behaviors in People with Intellectual Disabilities to Reduce Challenging Behaviors and Inappropriate Psychotropic Drug Prescribing Compared with Care as Usual: A Cluster-Randomized Trial"

_ijerph, 2024, doi:10.3390/ijerph21070950_

Round 1
Reviewer 1 Report
Comments and Suggestions for Authors
Dear authors,
Compliments for a very well written paper. It is very clear that the authors have a broad view over this field of research. It is clear that although a lot of effort was done to recruit participants, due to the corona pandemic and high working load of personnel, the intended sample could not be included.
All aspects of the RCT were attended to: pre-registration, examining the sample power, ethical issues, etc. Compliments for the way this was conducted.
I have a few minor points:
- Although the authors are transparent about the results, to my opinion they really can be more positive about the reduction of psychotropic drug prescriptions. It seems that the result of no change in behaviour overshadows this significant outcome.
- In 2.6 the sample size is described in great extent. You wanted to include 106 participants. However, 33 participant data was analyzed But, at no other point in the paper de ‘power’ was mentioned again. I think it is important to reflect on this in the discussion of the paper.
- One area of knowledge you didn’t cover was the area discussed by Barrett et al. 2023.
Impact of the emotional development approach on psychotropic medication in adults with intellectual and developmental disabilities: a retrospective clinical analysis. https://onlinelibrary.wiley.com/doi/10.1111/jir.13136
Could it be that there was no change in behaviour as in the intervention no, or little, attention was given to the emotional needs of the participants. In other words, the intervention is so ‘structured’ that too little attention is given to the emotional functioning of the participant.
Some minor errors:
Abstract: '..while CB's did not increase'. But, in the title you mention that you focus on 'to reduce CB's'. This is not in line with each other.
See When referring to the tables use a capital letter "Table X'
page 16: 4.2 - ....integrative care in in this... = integrative care in this....
Author Response
Comments and Suggestions for Authors
Dear authors,
Compliments for a very well written paper. It is very clear that the authors have a broad view over this field of research. It is clear that although a lot of effort was done to recruit participants, due to the corona pandemic and high working load of personnel, the intended sample could not be included.
All aspects of the RCT were attended to: pre-registration, examining the sample power, ethical issues, etc. Compliments for the way this was conducted.
Answer:
We thank the referee for the compliments.
I have a few minor points:
1) Although the authors are transparent about the results, to my opinion they really can be more positive about the reduction of psychotropic drug prescriptions. It seems that the result of no change in behaviour overshadows this significant outcome.
Answer:
While it is true that there was a significant decrease in psychotropic medication dosage from baseline to week 40 in the intervention group (as shown in Table 3), there was no significant effect of the intervention on psychotropic medication dosage in analyzes that also took the control group into account. Obviously, the sample size plays a role here. Therefore, we added a sentence to the limitations section of the Discussion:
‘Especially a possible effect of the intervention on the reduction of the psychotropic drug dose may have remained undetected due to lack of power caused by the small number of participants and the incomplete course of many interventions.’
2) In 2.6 the sample size is described in great extent. You wanted to include 106 participants. However, 33 participant data was analyzed But, at no other point in the paper de ‘power’ was mentioned again. I think it is important to reflect on this in the discussion of the paper.
Answer:
We agree with the referee and have added a phrase to the first sentence of the Limitations section in the Discussion:
‘The main limitation of this randomized controlled study was the small sample size and subsequent lack of power to test our study hypothesis of decreasing challenging behaviors and reducing psychotropic drug use.’
3) One area of knowledge you didn’t cover was the area discussed by Barrett et al. 2023.
Impact of the emotional development approach on psychotropic medication in adults with intellectual and developmental disabilities: a retrospective clinical analysis. https://onlinelibrary.wiley.com/doi/10.1111/jir.13136
Could it be that there was no change in behavior as in the intervention no, or little, attention was given to the emotional needs of the participants. In other words, the intervention is so ‘structured’ that too little attention is given to the emotional functioning of the participant.
Answer:
We agree that emotional health and support needs are very important. They were explicitly included in our integrative care intervention, yet we realize that not explicitly mentioning them in our manuscript in the Methods section has caused confusion. We have now added the word ‘emotional’ when describing the aim of the integrative care intervention, Methods section 2.3:
‘The intervention was aimed at assessing and addressing participants’ mental, emotional, and physical health - and support needs in order to optimize their functioning.’
We also added two sentences to the description of the intervention and the reference to Barret et al. (2024) in the Method section to emphasize the importance of emotional development in relation to challenging behavior and inappropriate psychotropic drug use in persons with intellectual disability:
‘Attention was also paid to the emotional development of the person to better understand the nature of the challenging behavior and to tailor the daily guidance to the person’s emotional needs. As Barret et al. (2024) showed, this approach can also be helpful in the reduction of inappropriate psychotropic drug prescriptions [29].’
Comments on the Quality of English Language
Some minor errors:
4) Abstract: '..while CB's did not increase'. But, in the title you mention that you focus on 'to reduce CB's'. This is not in line with each other.
Answer:
We agree with the referee and changed the wording into:
’ In the intervention group, however, the psychotropic drug dose decreased significantly while CBs did not change.’
5) See When referring to the tables use a capital letter "Table X'
Answer:
We apologize for overlooking this when proofreading and have changed the lowercase in uppercase where applicable.
6) page 16: 4.2 - ....integrative care in in this... = integrative care in this....
Answer:
We have removed the typo.
Reviewer 2 Report
Comments and Suggestions for Authors
very interesting topic
It could be useful to describe more specific the different treatment used in the integrative treatment;
a network model is used (as multistep multinetwork model o similar) ?
Author Response
Comments and Suggestions for Authors
very interesting topic
Answer.
We thank the referee for this remark.
1) It could be useful to describe more specific the different treatment used in the integrative treatment;
Answer:
We thank the referee for this suggestion. Next to the treatment interventions that are described in the Methods section 2.3 we added some other specific treatments:
‘The next step was to write an intervention/treatment plan, after 6 weeks at most. The interventions within that integrative treatment plan included additional diagnostics, individual medical, paramedic, pharmacological (including deprescribing), trauma, psychomotor and behavioral therapy, and other psychosocial, body-oriented and systemic/environmental interventions. Treatment was based on the outcomes of the diagnostic phase and was applied for the major problems identified by the specialized mental healthcare team during that phase.’
2) A network model is used (as multistep multinetwork model or similar) ?
Answer:
We assume that the referee means the framework developed by Shankar et al. (2019). As we described in the Methods section 2.3 this framework includes seven domains which should be addressed regarding diagnostics and treatments. The first phase of the intervention was the diagnostic phase of 6-8 weeks followed by the treatment phase of 30-32 weeks. In those terms the framework is a two-step model. However, additional diagnostics could be applied in the next phase, so the phases could overlap. With regard to addressing the different domains, diagnostic and treatment interventions could take place simultaneously. We added a clearer explanation of this operationalization of the framework of Shankar et al. (2019) to section 2.3:
- During the subsequent treatment phase of 30-32 weeks..
- Depending on the participants’ condition and the number, nature and urgency of the interventions, additional diagnostics and treatments could be applied simultaneously or sequentially.
Reviewer 3 Report
Comments and Suggestions for Authors
An interesting and valueble study, unfortunately with a really small sample size. The article is nicely written and logical but the presentation of results can be improved. I believe it deserves to be published with minor revision.
Comments:
Abstract:
-please avoid abbreviations in the abstract
Introduction:
P1 , l 36 – you need a reference for the statement.
Methods:
2.1.1. Recruitment – what was the expected population? How many people with ID are in the region approximately?
Statistical anaylses: as the sample was small , where the assumptions for linear regression met? Please replace linear regression with logistic regression or avoid presenting regresison at all.
Results:
It is not clear how 44 participants were sleceted for eligbility if you have calculated that you need 106 participants.
Figure 1 is not completely clear. The last arm – anaysis – did you have 15 and 18 participants or 12 and 16?
p.7, l 296-301 – this is a text for discussion (limitations of the study)
table 1 – has to be revised, pelase present data separate for each group and for total, please state number (n) everywhere and avoid percentage as the n is very small write n. Insert p values for comparisons you mention in the text (l 340-349)
p 11, l 390 – please delete this
l 371-389 belong to the discussion
Table 3, please insert p values. If you have used wilcoxon sign test then you should present average values as median and 25-75 percentiles.
Discussion
Well written and pertinent.
Author Response
Comments and Suggestions for Authors
An interesting and valuable study, unfortunately with a really small sample size. The article is nicely written and logical, but the presentation of results can be improved. I believe it deserves to be published with minor revision.
Answer:
We thank the referee for this compliment. We have considered referee’s comments and addressed these as much as possible. We changed the manuscript accordingly and think by doing so, the quality of the presentation of the results and the manuscript has improved.
Comments:
1) Abstract:
-please avoid abbreviations in the abstract.
Answer:
We agree with the referee that it is better not to use abbreviations in an abstract of a paper. However, given the limited word count, we had to abbreviate the words ‘Challenging Behavior’ and ‘Intellectual Disability’, into their well-known abbreviations (CB and ID), because otherwise we could not provide the necessary information in the abstract.
2) Introduction:
P1 , l 36 – you need a reference for the statement.
Answer:
We added the reference of Emerson (1953): These behaviors may include aggression (towards self, other persons, or materials), withdrawn and disruptive behaviors, hyperactivity, and irritability [4].
3) Methods:
2.1.1. Recruitment – what was the expected population? How many people with ID are in the region approximately?
Answer:
Unfortunately, we do not have recent data on this issue. In the Netherlands there are approximately 68,000 people with a moderate, severe, or profound intellectual disability (Ras, Verbeek-Oudijk & Eggink, 2013). Most of those people live in institutions, which are not equally spread over the Netherlands. Recruitment was complex, because we could not predict how many institutions in the north-east and middle of the Netherlands would participate in the project and if so, how many units of participating institutions were available. However, we assumed that we could recruit 8 institutions with 6-8 participating units/clusters each. We described the calculation of the sample size and number of clusters in the Method section 2.6 and added a sentence with the number of institutions and clusters to be recruited:
‘We assumed that we could recruit 8 institutions with 6-8 participating clusters (with on average 5 eligible persons) each in our catchment area.’
4) Statistical anaylyses: as the sample was small, where the assumptions for linear regression met? Please replace linear regression with logistic regression or avoid presenting regression at all.
Answer:
We agree with the referee that the sample size may be too small for linear regression analyzes. Also, assumptions were not fully met for all variables we used in the linear regression analyzes. Therefore, we decided to follow the referee’s suggestion to avoid presenting the results of the regression analyzes and removed the sentences regarding regression analyses in the Method section 2.7. Statistical analyses (Page 6, Line 271-275) and the results of these analyses in the Results section 3.4 (Page 11, Line 406-410) of the original submission.
5) Results:
It is not clear how 44 participants were selected for eligibility if you have calculated that you need 106 participants.
Answer: We apologize for this confusion. As will be clear from the manuscript, we were not able to include the number that we aimed for. Adding information on how we assumed to reach the 106 participants needed for sufficient power in the Methods section (point 3 of this referee) emphasizes our aim. Additionally, we added information to the results section to better explain the results of the recruitment process.
‘We recruited six care providing institutions for participation in the study, which subsequently selected 23 participating clusters. From the clusters the affiliated clinicians selected in total 44 eligible participants. For 38 of those 44 eligible participants informed consent was provided by their legal representatives. Of the 38 participants included in the study 18 were randomly allocated to the intervention group spread over 11 clusters and 20 to the control group spread over 12 clusters.’
Last, we now address the difference between the aimed number and the reached number of participants in the Discussion.
‘The sample size was much smaller than anticipated. We recruited 38 participants and were able to analyze data of 33 participants, where we expected to include three times as many, based on the catchment area (expected n of eligible clients was 240-320; attrition rate 50%). We expected the calculated sample size to be reachable, by including eight institutions, with an average of six to eight clusters and about four to six eligible participants per cluster. However, recruitment was heavily hindered by organizational and personnel problems of the service providers and organizational problems resulting from COVID restrictions.’
6) Figure 1 is not completely clear. The last arm – analysis – did you have 15 and 18 participants or 12 and 16?.
Answer:
We apologize for this confusion. We removed the information about the care provider who had withdrawn from the last arm Analysis; this information is written in the arm Allocation
7) p.7, l 296-301 – this is a text for discussion (limitations of the study).
Answer:
These lines describe factual results of the study and therefore we prefer to keep them in the Results section. In the discussion we elaborate on this issue, in the section 4.1. Study limitations.
‘The main limitation of this randomized controlled study is the small sample size.’
8) table 1 – has to be revised, please present data separate for each group and for total, please state number (n) everywhere and avoid percentage as the n is very small write n. Insert p values for comparisons you mention in the text (l 340-349).
Answer:
We thank the referee for these useful suggestions and have added two columns showing the data of each group separately and the number of participants in addition to the percentages.
The p-values and other test statistics mentioned in Line 340-349 (original submission) are now added to Table 3.
9) p 11, l 390 – please delete this.
Answer.
The lay-out at this place may be somewhat confusing. It seems that the first sentence is a doubling with the sentence below, However, the first sentence is a sentence in the main text, referring to Table 2. The second sentence is the Table title and explanation of Table 2. We moved the first sentence referring to Table 2 in the main text to Page 11/line 364.
10) l 371-389 belong to the discussion.
Here, we report about the results of a process in which we applied the framework of Shankar et al. The aim of our study was to investigate the results of the multidisciplinary and integrative care program. In our opinion, reporting on the way the program was carried out is part of the results section.
11) Table 3, please insert p values. If you have used wilcoxon sign test then you should present average values as median and 25-75 percentiles.
Answer:
We changed Table 3 by replacing mean and SD by median and IQR. Also, we added p=values.
12) Discussion
Well written and pertinent.
Answer:
We thank the referee for this remark.